# Agro-Physiological Indices and Multidimensional Analyses for Detecting Heat Tolerance in Wheat Genotypes

**Ibrahim Al-Ashkar** [1,2,*], **Mohammed Sallam** [1], **Abdelhalim Ghazy** [1], **Abdullah Ibrahim** [1], **Majed Alotaibi** [1], **Najeeb Ullah** [3] **and Abdullah Al-Doss** [1]

1 Department of Plant Production, College of Food and Agriculture Sciences, King Saud University, Riyadh 11451, Saudi Arabia
2 Agronomy Department, Faculty of Agriculture, Al-Azhar University, Cairo 11651, Egypt
3 Queensland Alliance for Agriculture and Food Innovation, The University of Queensland, St Lucia, QLD 4072, Australia
* Correspondence: ialashkar@ksu.edu.sa

**Abstract:** Increasing atmospheric temperature can significantly reduce global wheat productivity; despite a mounting demand for wheat grain supplies. Developing genotypes with superior performance under current and future hot climates is a key challenge for wheat breeders. Multidimensional tools have supported plant breeders in increasing the genetic stability rate of agro-physiological indices that influence wheat productivity. We used 25 agro-physiological indices to classify 20 bread wheat genotypes for their heat stress tolerance. Agro-physiological indices and multidimensional analyses to identify differences in genetic and phenotypic were used, combining these analyses to reach selection criteria of accurate and credible. The 25 studied indices reflected high genotypic and environmental variations. We used 16 indices, which have brought together high heritability and genetic gain as indicators for screening heat-tolerant genotypes. Based on the seven principal comprehensive indices of (D value), wheat genotypes were classified into three highly heat-tolerant, four heat-tolerant, six moderately heat-tolerant, five heat-sensitive, and two highly heat-sensitive wheat genotypes. Based on four critical indices [grain yield (GY), grain-filling duration (GFD), spike length (SL) and canopy temperature (CT)] obtained from stepwise multiple linear regression (SMLR), the genotypes were grouped as four genotypes highly heat-tolerant, six heat-tolerant, two moderately heat-tolerant, four heat-sensitive and four highly heat-sensitive. The classification D value and SMLR distances were significantly correlated based on the Mantel test, with a perfect match in nine genotypes. SMLR indicated that a mathematical equation for the evaluation of wheat heat tolerance was established: $GY = 0.670 + 0.504 \times GFD + 0.334 \times SL - 0.466 \times CT$ ($R^2 = 0.739$; average prediction accuracy of 94.12%). SMLR-based classification of wheat genotypes for heat tolerance was further verified through discriminant analysis, which showed that prior and posterior classification was identical in eighteen genotypes. Cross-validation showed that prior and posterior classification was identical in thirteen genotypes. Based on this study, we can recommend tolerated new wheat lines (DHL25, DHL05, DHL23 and DHL08) and cultivar Pavone-76 as a promising genetic source for heat-tolerant breeding programs.

**Keywords:** heat stress; bread wheat; agro-physiological indices; genetic parameters; multidimensional analysis; cross-validation

## 1. Introduction

Agriculture is the fundamental economic sector in the Arab region, providing employment, food security, and income for residents of rural areas. The last few decades have witnessed a cascading impact of climate change, resulting in a rapid shift in natural ecosystems, agricultural productivity, and cultivation practices. Generally, the wheat crop is adapted to a broad range of global climatic conditions. However, wheat cropping systems

have started a marginal decline in grain yield, primarily because of groundwater depletion, land degradation, and heat stress. In contrast, wheat demand has gradually increased, i.e., a 2% increase in annual wheat production is needed to feed the growing world population [1]. It is projected that wheat grain yield must be increased by 60% compared to today to meet the supplies by 2050 [2]. A significant rise in the atmospheric temperature is one of the most significant challenges for wheat producers around the globe as it seriously affects grain yield and quality [2–6].

Wheat crops have a temperature requirement of 12–22 °C for most developmental stages, notably during flowering and grain filling [7]. Heat stress constitutes one of the main threats to sustainable wheat production. It is even more detrimental than drought as it seriously impairs metabolic processes in all plant organs during most of the developmental stages of the crop [8]. Late-cultivated wheat crops suffer from high canopy temperatures (>31 °C) from anthesis to maturity, which negatively affects grain yield formation [1,9]. In wheat, a 3–4 °C rise in the seasonal minimum or maximum temperatures could decrease yield by 15–35% in Africa and Asia and by 25–35% in the Middle East across the flowering, pollination, and grain-filling periods [1,10,11]. Asseng, et al. [12] and Djanaguiraman et al. [8] have projected a 6% reduction in global wheat yield for each 1 °C increase in the average current temperature. This impact will become more variable or extreme through times and places. When such stress occurs during sensitive growth stages, such as flowering or grain filling, yields are likely to be much reduced, especially with short episodes of very high temperatures [4]. Heat stress significantly reduces leaf photosynthesis, arresting overall plant growth and yield [13] by limiting carbohydrate formation and supplies. A temperature above the optimal range damages the leaf thylakoid membrane, reduces photosynthetic electron transfer, and arrests photochemical reactions [8,13–15].

High temperature during anthesis reduces grain number per spike by restricting embryo development and inducing pollen sterility [1,16]. Post-anthesis (i.e., during the grain-filling period) heat reduces grain-filling rate and duration by accelerating leaf senescence [7,16,17]. Researchers often exploit multiple physiological mechanisms for improving wheat performance in hot climates. Physiological traits such as early maturity, minimization of canopy temperature, stay green, and high stem water-soluble carbohydrates accumulation, coupled with high biomass accumulation, are likely to contribute to final grain yield [1,7,9,17–19]. Early maturity is considered one of the main physiological responses of wheat genotypes to high temperatures. This accelerates different physiological mechanisms to contribute to grain yield production, although there was a negative correlation between them [1]. Stay green trait is the principal source of chlorophyll contents in the leaf and is considered important with a direct impact on grain yield and its components, but the green area decreases under heat stress [17,20]. Likely, the plant may compensate for the negative impact of heat-induced green area loss on final grain size by remobilizing the stem water-soluble carbohydrates stockpiled [7,21]. Some wheat genotypes have the capacity to cool the canopy during grain filling, which can access underground moisture, thereby promoting evaporation and photosynthesis conservation under hot conditions [22]. Low canopy temperature positively correlates with sustained grain yield formation under hot environments in wheat genotypes [7,23,24].

Accordingly, these physiological traits could be used as an indirect selection tool for the genetic improvement of wheat genotypes; and they are less affected by the environment and have more genetic stability compared with grain yield traits [1,22,25]. Further exploration of physiological traits is needed for detecting wheat genotypes that produce high yields and adapt to high temperatures using these traits as selection criteria. This could only be achieved by developing high-yielding, early maturing genotypes that have protracted grain-filling duration, climate-smart, and abiotic stress tolerance [7,18]. Given the global climate changes, these efforts should be promoted to counter the negative impacts of high temperatures [26,27]. Efforts have been made to evaluate and test elite wheat genotypes for their adaptability to heat and traits-linked, combined into typical varieties for both high-yielding and heat tolerance [2,9,18]. Evaluation of the genetic parameters for agro-

physiological response is important to determine the best genotypes used in breeding programs to tolerate abiotic stress [1].

The application of multivariate analysis techniques and multidimensional methods to achieve accuracy in verification and selection supports breeding programs and their success by incorporating agro-physiological traits. This may provide a better understanding, and an integrative approach that combines various parameters of heat adaptive traits and their integration into breeding programs may help show desirable genotypes [6,28–30]. The use of a combination of analyses is needed for reliably studying phenotypic traits generated through high computing-powered modeling of multidimensional data and to provide a more comprehensive understanding of the complicated mechanisms under heat stress [28,31,32]. Therefore, multivariate analysis techniques [principal component analysis (PCA), multiple regression, path analysis, cluster analysis, multivariate analysis of variance (MANOVA), and discriminant analysis] can serve as a model instrument for screening tests and for separating sources of variation [29,33,34].

For instance, PCA reduces the number of variables of a data set and uses an orthogonal transformation to convert a set of observations of possibly correlated variables into a set of values of uncorrelated variables while preserving as much information as possible. Multiple regression is used to determine the model's overall fit (variance explained) and the relative contribution of each variable to the total variance explained. Path analysis is a straightforward extension of multiple regression [1]. It aims to provide estimates of the magnitude and significance of hypothesized causal connections between sets of variables and divide them into direct and indirect impacts. Cluster analysis is a method for collecting similar genotypes into several clusters based on the values of several variables used. Discriminant analysis and MANOVA are used to assess the extent of matching a classification by prior knowledge of the membership of the genotype within each group and to predict the classification position of new cases that have not been classified [29,35]. The main aims of this study were (i) to develop a screening method to identify the contribution of the key heat tolerance indices of wheat using multidimensional evaluation (ii) to evaluate and cluster the heat tolerance of 20 wheat genotypes (iii) to verify the validity of classification and predicting of new cases that have not been classified. The results presented here would be significant for further screening and evaluation of heat tolerance wheat and for improving wheat breeding that can be utilized to breed heat-tolerant varieties.

## 2. Materials and Methods

### 2.1. Experimental Design and Plant Materials

Field trials were conducted at the King Saud University Agricultural Research Station (24°42′ N, 44°46′ E, 400 m asl) for two successive years (2018/2019 and 2019/2020). The experimental design was a randomized complete block with three independent replicates. Two successive experiments were performed each year, i.e., the optimum sowing (15 and 20 November in 2018 and 2019, respectively) and late sowing (15 and 20 December in 2018 and 19, respectively). The late-sown crops experienced an average temperature of 30.4–31.0/14.2–14.4 °C day/night during grain-filling duration in the two seasons. The experimental unit (plot) consisted of five rows (3.0 m long) each, with the distance between rows of 0.17 m, at a seedling rate of 360 germinating kernels $m^{-2}$, and the fertilizing rates used were180 kg $ha^{-1}$ of N and 31 kg $ha^{-1}$ of $P_2O_5$. The electrical conductivity of soil texture (2.89 dS $m^{-1}$). The temperatures during the growing season are presented in Table S1. Twenty different wheat genotypes (6 varieties and 14 lines) were used in this study. The six varieties were chosen based on the presence of broad genetic variations between them concerning tolerance and/or sensitivity to heat, which were supplied by the Agricultural Research Center, Egypt (Table S2). The Department of Agronomy supplied the 14 doubled haploid lines (DHLs), Faculty of Agriculture, Al-Azhar University, Cairo, Egypt, and previously published [36] and been used to assess heat tolerance (Table S2).

### 2.2. Measurements of Traits and Data Collection

All 20 wheat genotypes were assessed using 25 physio-morphological and agronomic attributes. The mean value of five samples and/or plants per genotype for measuring all physio-morphological and some agronomic attributes were randomly selected from middle rows to reduce environmental impact and repeated three times. Twelve physiological crop attributes were estimated at the grain-filling stage. These include canopy temperature (CT), leaf water content (LWC), relative water content (RWC), photosynthesis rate (Pn), transpiration rate (E), stomatal conductance (Gs), intracellular $CO_2$ concentration (Ci), stomatal limitation value (Ls), peroxidase (POD), polyphenol oxidase (PPO), and catalase (CAT). Briefly, CT was measured using an infrared thermometer (Therma CAM SC 3000 infrared camera, FLIR System, USA). LWC and RWC were obtained by fresh weight (FW), turgid weight (TW), and dry weight (DW) after oven drying at 48 h at 70 °C [37,38] deceptively. Pn, E, Gs, and Ci were measured on the upper third of flag leaves by a Li-6400 gas exchange system (Li-Cor, Inc., Lincoln, NE, USA) from 10:00 AM until 12 noon.

The stomatal limitation value (Ls) was calculated from the following equation [39]; Ls = (1 − Ci)/ atmospheric $CO_2$ concentration

Antioxidant enzymes such as CAT, POD, and PPO were assessed from 0.5 g of fresh leaf samples. The leaves were ground in liquid nitrogen and placed in an ice bath with 50 mM potassium phosphate buffer (pH 7.8), including 1% (*w/v*) polyvinylpolypyrrolidone, and then centrifuged at 14,000× *g* rpm for 10 min at 4 °C for enzyme extractions. The supernatant is used as an extract of assays for CAT, POD and PPO, as described by Aebi [40], Chance and Maehly [41], and Duckworth and Coleman [42], respectively.

Four morphological attributes, i.e., green leaf number (GLN), flag leaf area (FLA), green leaf area (GLA), and leaf area index (LAI), were estimated after the anthesis. FLA and GLA were estimated using a leaf area meter (LI 3100; LI-COR Inc., Lincoln, NE, USA) after separating all the green leaves from the stem. LAI was calculated by the following formula: LAI = green leaf area / ground cover.

Ten agronomic attributes were estimated before and/or after harvest, namely the days to heading (DH, days), days to maturity (DM, days), grain-filling duration (GFD, days), plant height (PH, cm plant$^{-1}$), spike length (SL, after excluding awns), number of spikes (NS, m$^{-2}$), number of spikelets (NSS, spike$^{-1}$), number of kernels (NKS, spike$^{-1}$), thousand kernel weight (HKW, g) and grain yield (GY, ton ha$^{-1}$). DH was recorded upon reaching the 50% of plants to flowered, and DH was recorded upon reaching the 50% of plants to peduncles yellow. GFD was calculated from the period between DM and DH. After harvesting, the plants were threshed to measure NKS, HKW and GY traits. GY was estimated from three rows to two-meters long.

### 2.3. Statistical Analysis of Evaluated Data

Based on the foregoing attributes data, the heat tolerance indices (HTI) of each of the above attributes were calculated.

$$\text{HTI}_{ij} = \frac{x_{ij}\text{-under heat stress condition}}{x_{ij}\text{-under optimum condition}} \qquad (1)$$

where $\text{HTI}_{ij}$ is the heat tolerance index (*j*) for genotype (*i*); $x_{ij}$-under heat stress condition and $x_{ij}$-under optimum condition represent values of the index for the genotype evaluated under optimum condition and heat treatments, respectively.

Analysis of variance for 26 indices was implemented using SAS v9.2 software (SAS Institute, Inc., Cary, NC, USA) for each season. The combined analysis was implemented for the two-season using Bartlett's test to test the homogeneity of error variance between seasons. Afterwards, the variance of the combined data of all indices was used for calculating variance components and genetic parameters such as heritability (h$^2$, broad sense), genetic gain (%), genotypic coefficient of variability (GCV), and phenotypic coefficient of variability (PCV), following t Fehr [43] and described by Al-Ashkar et al. [29]. Multidimensional modeling was used to understand genotype × environment interactions and their contributions

of the key indices to genotypic performance. The principal component analysis (PCA) was performed based on data provided components (eigenvalue > 1) with the normalization values of heat tolerance indices used in calculating comprehensive indices values. It was also used for calculating the membership function value of heat tolerance. It was assessed using multiple indices as the subordinate function value (μ) and provides a comprehensive assessment using subordinate functions based on the theory of fuzzy mathematics [44]. The subordinate function μ was calculated using Equation (2).

$$\mu = \frac{x_i - x_{min}}{x_{max} - x_{min}} \tag{2}$$

where $x_i$ indicates the *i*th comprehensive index; $x_{max}$ and $x_{min}$ indicate the maximum and minimum value for the ith comprehensive index of all the genotypes, respectively.

In Equation (3), the weight function $W_i$ was calculated and represents the relative importance of the *i*th comprehensive index for a genotype; $P_i$ represents the contribution of the *i*th comprehensive index.

$$W_i = \frac{P_i}{\sum_{i=1}^{n} P_i \,,i=1,\,2\,,\,3.........n} \tag{3}$$

In Equation (4), the comprehensive index value for heat tolerance (D value) was calculated separately for each genotype to detect their heat tolerance, which was used for making clusters and evaluating heat tolerance.

$$D = \sum_{i=1}^{n} [\mu\,(x_i) * W_i] i = 1,\,2\,,\,3.........n \tag{4}$$

Stepwise multiple linear regression (SMLRA) and path analyses were used to identify the main indices and their contribution, respectively, to confirm the accuracy and reliability of clusters. The effective indices used in the calculation of cluster analysis (CA), building upon the genetic dissimilarity matrix between genotypes (Euclidean distance and Ward's method of agglomeration) in five major groups to tolerance. The discriminant function analysis (DFA) was used to reaffirm the classification of each genotype by analyzing main indices (as quantitative variables) with the five major groups (as qualitative variables). Statistical analysis (PCA, SMLRA, path analysis, CA, DFA, and multicollinearity test) carried out by XLSTAT statistical package software (vers. 2019.1, Excel Add-ins soft SARL, New York, NY, USA).

## 3. Results

### 3.1. Phenotypic Analysis of Heat Tolerance Index

3.1.1. Analysis of Variance and Genetic Parameters of the Studied Indices

Analysis of heat tolerance index data (Table 1) revealed highly significant ($p < 0.01$) genotypic variations for all studied indices across the two seasons as sources of variance differed across the genotypes under optimal and heat-stressed environments for all measured traits. Across the combined data of two seasons, the interaction was significant for nineteen indices out of twenty-five; and non-significant for seven indices (DH, MD, GFD, POD, PPO, CAT, and GY). Test of the homogeneity of error variance of heat tolerance index between two seasons showed non-significant for twenty-two indices, but error variance was heterogeneity for LAI, Gs, and GY. The broad-sense heritability ($h^2$) showed high values (>60%) for seventeen measured indices, which varied from 60.43% to 89.51% (Table 1). The genetic gain (GG) showed high values (>20%) for nine measured indices, which varied from 20.05% (GFD) to 54.25% (Ls); and moderate values (>10%) for twelve measured indices, which varied from 10.15% (DH) to 17.22% (Ci). The PCV and GCV were convergent for most indices, and the PCV was larger relative to the GCV, except for three indices (GLN, NSS, and NKS), which showed PCV values greater than GCV values twice.

**Table 1.** Analysis of variance for heat tolerance index of measured traits, heritability, genetic gain, genotypic, and phenotypic coefficients of variability for traits of 20 wheat genotypes.

| Source of Variations | | df | CT | LWC | RWC | Pn | Gs | Ci | E | Ls | POD | PPO | CAT | GLN | FLA |
|---|---|---|---|---|---|---|---|---|---|---|---|---|---|---|---|
| Season 1 | Replications | 2 | 0.0023 | 0.0018 | 0.0017 | 0.0018 | 0.0012 | 0.0012 | 0.0013 | 0.0031 | 0.0030 | 0.0014 | 0.0028 | 0.0015 | 0.0016 |
| | Genotypes (G) | 19 | **0.0104** | **0.0135** | **0.0092** | **0.0307** | **0.0797** | **0.0171** | **0.0449** | **0.4274** | **1.6312** | **0.5980** | **1.9533** | **0.0154** | **0.0103** |
| | Error | 38 | 0.0015 | 0.0014 | 0.0013 | 0.0013 | 0.0008 | 0.0010 | 0.0009 | 0.0025 | 0.0019 | 0.0012 | 0.0023 | 0.0013 | 0.0012 |
| Season 2 | Replications | 2 | 0.0028 | 0.0017 | 0.0017 | 0.0017 | 0.0014 | 0.0013 | 0.0013 | 0.0031 | 0.0029 | 0.0014 | 0.0027 | 0.0017 | 0.0014 |
| | Genotypes (G) | 19 | **0.0346** | **0.0080** | **0.0099** | **0.0424** | **0.1201** | **0.0212** | **0.0689** | **0.2249** | **1.6030** | **0.5882** | **1.9203** | **0.0445** | **0.0368** |
| | Error | 38 | 0.0026 | 0.0013 | 0.0012 | 0.0014 | 0.0015 | 0.0010 | 0.0010 | 0.0023 | 0.0019 | 0.0012 | 0.0023 | 0.0014 | 0.0011 |
| Combined | Seasons (S) | 1 | 0.1535 | 0.0533 | 0.0319 | 0.0218 | 0.0355 | 0.0041 | 0.0028 | 0.0490 | 0.0012 | 0.0010 | 0.0010 | 0.0908 | 0.0618 |
| | Replications (Sea.) | 4 | 0.0026 | 0.0018 | 0.0017 | 0.0018 | 0.0013 | 0.0013 | 0.0013 | 0.0031 | 0.0029 | 0.0014 | 0.0028 | 0.0016 | 0.0015 |
| | Genotypes (G) | 19 | **0.0368** | **0.0178** | **0.0140** | **0.0660** | **0.1915** | **0.0363** | **0.1036** | **0.6125** | **0.8210** | **0.9820** | **0.8670** | **0.0340** | **0.0369** |
| | S ×G | 19 | **0.0085** | **0.0041** | **0.0052** | **0.0072** | **0.0080** | **0.0018** | **0.0100** | **0.0410** | 0.0002 | 0.0001 | 0.0001 | **0.0261** | **0.0102** |
| | Error | 76 | 0.0020 | 0.0014 | 0.0013 | 0.0013 | 0.0012 | 0.0010 | 0.0009 | 0.0024 | 0.0019 | 0.0012 | 0.0023 | 0.0013 | 0.0012 |
| Heritability ($h^2$ %) | | | 64.53 | 60.43 | 60.61 | 80.10 | 89.51 | 77.94 | 84.43 | 86.66 | 62.05 | 83.66 | 85.61 | 27.50 | 61.88 |
| Genetic gain (GG %) | | | 14.95 | 10.95 | 10.87 | 22.05 | 43.21 | 17.22 | 31.12 | 54.25 | 50.44 | 37.60 | 44.10 | 5.76 | 13.78 |
| G.C.V. % | | | 9.03 | 6.83 | 6.78 | 11.96 | 22.17 | 9.47 | 16.44 | 28.29 | 31.08 | 19.96 | 23.14 | 5.33 | 8.50 |
| Ph.C.V. % | | | 11.24 | 8.79 | 8.71 | 13.36 | 23.43 | 10.73 | 17.89 | 30.39 | 39.46 | 21.82 | 25.00 | 10.17 | 10.81 |

| Source of variations | | df | GLA | LAI | DH | MD | GFD | NS | PH | SL | NSS | NKS | HW | GY |
|---|---|---|---|---|---|---|---|---|---|---|---|---|---|---|
| Season 1 | Replications | 2 | 0.0016 | 0.0005 | 0.0018 | 0.0017 | 0.0016 | 0.0012 | 0.0015 | 0.0015 | 0.0013 | 0.0016 | 0.0013 | 0.0012 |
| | Genotypes (G) | 19 | **0.0283** | **0.0441** | **0.0083** | **0.0062** | **0.0101** | **0.0299** | **0.0062** | **0.0240** | **0.0227** | **0.0139** | **0.0070** | **0.0260** |
| | Error | 38 | 0.0013 | 0.0004 | 0.0022 | 0.0013 | 0.0017 | 0.0009 | 0.0013 | 0.0013 | 0.0012 | 0.0012 | 0.0010 | 0.0013 |
| Season 2 | Replications | 2 | 0.0011 | 0.0008 | 0.0017 | 0.0015 | 0.0013 | 0.0011 | 0.0016 | 0.0034 | 0.0018 | 0.0016 | 0.0014 | 0.0008 |
| | Genotypes (G) | 19 | **0.0593** | **0.0381** | **0.0081** | **0.0113** | **0.0438** | **0.0311** | **0.0177** | **0.1135** | **0.0135** | **0.0231** | **0.0090** | **0.0834** |
| | Error | 38 | 0.0010 | 0.0007 | 0.0014 | 0.0012 | 0.0010 | 0.0008 | 0.0013 | 0.0023 | 0.0012 | 0.0014 | 0.0011 | 0.0008 |
| Combined | Seasons (S) | 1 | 0.4502 | 0.1920 | 0.0001 | 0.0423 | 0.002 | 0.0214 | 0.0030 | 0.5210 | 0.0337 | 0.0636 | 0.0183 | 0.0560 |
| | Replications (Sea.) | 4 | 0.0013 | 0.0007 | 0.0017 | 0.0016 | 0.0014 | 0.0011 | 0.0016 | 0.0024 | 0.0015 | 0.0016 | 0.0013 | 0.0010 |
| | Genotypes (G) | 19 | **0.0582** | **0.0520** | **0.0154** | **0.0164** | **0.046** | **0.0414** | **0.0139** | **0.0501** | **0.0182** | **0.0244** | **0.0112** | **0.0816** |
| | S ×G | 19 | **0.0293** | **0.0304** | 0.0009 | 0.0001 | 0.0070 | **0.0196** | **0.0196** | 0.0100 | **0.0873** | **0.0183** | **0.0127** | **0.0047** | 0.0281 |
| | Error | 76 | 0.0011 | 0.0005 | 0.0018 | 0.0013 | 0.0013 | 0.0009 | 0.0013 | 0.0018 | 0.0012 | 0.0013 | 0.0011 | 0.0011 |
| Heritability ($h^2$ %) | | | 49.07 | 38.65 | 55.22 | 60.47 | 69.29 | 47.44 | 45.30 | 69.45 | 19.34 | 33.73 | 56.69 | 80.73 |
| Genetic gain (GG %) | | | 13.67 | 15.06 | 10.15 | 10.35 | 20.05 | 12.20 | 6.13 | 14.75 | 2.40 | 5.85 | 10.36 | 20.78 |
| G.C.V. % | | | 9.47 | 11.76 | 6.63 | 6.46 | 11.69 | 8.59 | 4.42 | 8.59 | 2.65 | 4.89 | 6.68 | 11.23 |
| Ph.C.V. % | | | 13.52 | 18.91 | 8.92 | 8.31 | 14.05 | 12.48 | 6.57 | 10.31 | 6.03 | 8.42 | 8.87 | 12.50 |

Values in bold indicate significance at *p* < 0.05.

3.1.2. Heat Tolerance Index of Measured Parameters

The heat tolerance index showed highly significant variations between genotypes as sources of variance due to changes between optimal and heat treatment for all measured indices. Interestingly, in S1, S2, and across the combined data of two seasons, there was a significant variation between large and small values for the majority of measured indices, and with certain exceptions for some indices (DH, MD, HW, and CT), which showed limited variation between large and small value. The large values were twice or more than the small ones for the seven measured indices (LAI, Gs, E, POD, PPO, CAT, and GY), a reference to the efficiency of these indices in the show genetic diversity between genotypes used (Table 2). However, five indices (CT, Pn, POD, PPO, and CAT) in some genotypes were higher (HTI > 1). Different indices within the same genotype and the same index within different genotypes showed considerable variability in the heat-tolerant index. Thus, it is difficult to make an accurate and reliable assessment of the heat tolerance of each genotype using the HTI of a single index. The values for coefficients of experimental variation (CVe) were small, ranging from 3.94% to 5.43.

*3.2. Multidimensional Analyses in the Classification of Heat-Tolerant Genotypes*

3.2.1. Principal Component Analysis

Kaiser–Meyer–Olkin measure of sampling adequacy showed a high value (KMO > 0.5). Based on the absolute value of each eigenvector, the PCA grouped the estimated variables (25 indices) into seven principal comprehensive indices (eigenvalue > 1), which had contribution rates of 35.47% (PCI1), 15.99% (PCI2), 10.88% (PCI3), 7.89% (PCI4), 5.20% (PCI5), 4.38% (PCI6), and 4.28% (PCI7). The cumulative contribution rate reached 84.09% (Table 3). PCI1 was related to DH, MD, GFD, NS, PH, GLN, LAF, LAI, SL, RWC, HW CT, Ls, PPO, CAT, and GY. PCI2 was related to Gs, Ci, and E. PCI3 was related to NSS and Pn; PCI4 was related to LWC and NKS; PCI5 and PCI6 were related to GLA and POD, respectively (Table 2). Figure 1 shows the correlations between variables (RWC and Ci) with factors (PCA1 and PCA2) were the highest value for squared cosines (0.885 and 0.759), respectively. So, the vectors were for RWC and Ci longer and acute with factors. As shown in Table 3, the seven principal comprehensive indices (PCI) were calculated for each genotype.

According to Equation (2), the subordinate function μ values were computed for comprehensive scores of each genotype. These values explain the heat tolerance of each genotype. Genotypic heat tolerance can be assessed based on the subordinate function μ value within the same comprehensive index. If the $\mu(x_i)$ is 1.000, then the genotype has the highest heat tolerance; if the $\mu(x_i)$ is 0.000, the genotype has the highest heat sensitivity. For PCI1, genotype DHL08 had the highest $\mu(x_1)$ of 1.000, which showed that it had the highest heat tolerance (Table 4). The DHL01 had the smallest $\mu(x_1)$ of 0.000, which showed that it had the highest heat sensitivity. However, in PCI2, genotype DHL02 had the highest $\mu(x_1)$ of 1.000, and KSU106 had the smallest $\mu(x_1)$ of 0.000. The evaluation result under each comprehensive index was aligned with our phenotypic results but differed from one PCI to another.

To maximize the effectiveness of comprehensive evaluation for the heat tolerance of used wheat genotypes was needed, the weight factor was computed for seven comprehensive indexes (W1: 0.422, W2: 0.190, W3: 0.129, W4: 0.094, W5: 0.062, W6: 0.052 and W7: 0.051) (Table 3) according to Equation (3) to maximize the effectiveness of the heat tolerance evaluation technique. A comprehensive evaluation value D was computed for heat tolerance (Table 4) according to Equation (4). The D value for twenty genotypes varied from 0.686 (HT) to 0.176 (HS). The cluster analysis from D values was then used to classify genotypes into five main clusters (Figure 2).

**Table 2.** Means ± standard deviation, ranges of the 20 genotypes of two seasons and their combined for 25 indices.

| Indices | S1 | | | S2 | | | Combined Data | | | |
|---|---|---|---|---|---|---|---|---|---|---|
| | Min | Max | Mean ± SD | Min | Max | Mean ± SD | Min | Max | Mean ± SD | CV |
| CT | 0.998 | 1.081 | 0.989 ± 0.059 | 0.950 | 1.131 | 0.988 ± 0.107 | 0.980 | 1.213 | 0.972 ± 0.045 | 4.60 |
| LWC | 0.804 | 0.974 | 0.955 ± 0.067 | 0.831 | 0.910 | 0.891 ± 0.051 | 0.826 | 0.962 | 0.913 ± 0.037 | 4.10 |
| RWC | 0.807 | 0.950 | 0.922 ± 0.055 | 0.769 | 0.891 | 0.889 ± 0.057 | 0.829 | 0.935 | 0.906 ± 0.036 | 3.98 |
| Pn | 0.700 | 0.987 | 0.895 ± 0.101 | 0.741 | 0.996 | 0.921 ± 0.118 | 0.721 | 1.050 | 0.908 ± 0.036 | 3.97 |
| Gs | 0.377 | 0.871 | 0.704 ± 0.163 | 0.386 | 0.881 | 0.738 ± 0.200 | 0.382 | 0.995 | 0.722 ± 0.035 | 4.80 |
| Ci | 0.657 | 0.917 | 0.795 ± 0.076 | 0.654 | 0.904 | 0.808 ± 0.084 | 0.656 | 0.877 | 0.802 ± 0.032 | 3.94 |
| E | 0.465 | 0.830 | 0.754 ± 0.122 | 0.433 | 0.850 | 0.764 ± 0.151 | 0.449 | 0.946 | 0.760 ± 0.030 | 3.95 |
| Ls | 0.745 | 0.981 | 0.932 ± 0.377 | 0.745 | 0.970 | 0.901 ± 0.273 | 0.745 | 0.959 | 0.903 ± 0.049 | 5.43 |
| POD | 0.072 | 1.146 | 0.832 ± 0.737 | 0.071 | 1.145 | 0.825 ± 0.731 | 0.072 | 1.154 | 0.829 ± 0.044 | 5.26 |
| PPO | 0.085 | 1.164 | 0.768 ± 0.446 | 0.084 | 1.162 | 0.762 ± 0.442 | 0.085 | 1.068 | 0.865 ± 0.035 | 4.00 |
| CAT | 0.103 | 1.153 | 0.915 ± 0.807 | 0.102 | 1.197 | 0.909 ± 0.800 | 0.103 | 1.627 | 0.912 ± 0.048 | 5.26 |
| GLN | 0.693 | 0.920 | 0.889 ± 0.072 | 0.705 | 0.930 | 0.904 ± 0.121 | 0.760 | 0.994 | 0.902 ± 0.036 | 4.00 |
| FLA | 0.687 | 0.928 | 0.884 ± 0.059 | 0.615 | 0.918 | 0.838 ± 0.110 | 0.703 | 0.920 | 0.861 ± 0.035 | 4.02 |
| GLA | 0.612 | 0.991 | 0.888 ± 0.097 | 0.449 | 0.970 | 0.796 ± 0.140 | 0.610 | 0.913 | 0.826 ± 0.033 | 4.02 |
| LAI | 0.329 | 0.712 | 0.471 ± 0.121 | 0.328 | 0.661 | 0.551 ± 0.112 | 0.345 | 0.671 | 0.511 ± 0.022 | 4.37 |
| DH | 0.838 | 0.948 | 0.835 ± 0.053 | 0.858 | 0.945 | 0.934 ± 0.052 | 0.848 | 0.978 | 0.870 ± 0.042 | 4.87 |
| MD | 0.848 | 0.911 | 0.812 ± 0.045 | 0.787 | 0.933 | 0.837 ± 0.061 | 0.817 | 0.952 | 0.845 ± 0.036 | 4.27 |
| GFD | 0.693 | 0.926 | 0.779 ± 0.058 | 0.575 | 0.958 | 0.769 ± 0.120 | 0.694 | 0.983 | 0.712 ± 0.036 | 5.06 |
| NS | 0.583 | 0.794 | 0.752 ± 0.100 | 0.559 | 0.809 | 0.726 ± 0.101 | 0.596 | 0.813 | 0.739 ± 0.030 | 4.06 |
| PH | 0.793 | 0.873 | 0.855 ± 0.046 | 0.736 | 0.854 | 0.804 ± 0.076 | 0.824 | 0.928 | 0.889 ± 0.036 | 4.05 |
| SL | 0.751 | 0.959 | 0.911 ± 0.089 | 0.770 | 0.947 | 0.907 ± 0.194 | 0.803 | 0.959 | 0.912 ± 0.042 | 4.65 |
| NSS | 0.664 | 0.892 | 0.851 ± 0.087 | 0.754 | 0.913 | 0.884 ± 0.067 | 0.760 | 0.881 | 0.857 ± 0.035 | 4.04 |
| NKS | 0.764 | 0.979 | 0.883 ± 0.068 | 0.756 | 0.980 | 0.927 ± 0.087 | 0.760 | 0.981 | 0.905 ± 0.036 | 3.98 |
| HW | 0.711 | 0.817 | 0.803 ± 0.048 | 0.720 | 0.845 | 0.828 ± 0.054 | 0.742 | 0.852 | 0.816 ± 0.033 | 4.06 |
| GY | 0.444 | 0.894 | 0.776 ± 0.092 | 0.436 | 0.851 | 0.704 ± 0.166 | 0.440 | 0.899 | 0.740 ± 0.033 | 4.48 |

Canopy temperature (CT), leaf water content (LWC), relative water content (RWC), photosynthesis rate (Pn), stomatal conductance (Gs), intracellular CO2 concentration (Ci), transpiration rate (E), stomatal limitation value (Ls), peroxidase (POD), polyphenol oxidase (PPO), catalase (CAT), green leaf number (GLN), flag leaf area (FLA), and green leaf area (GLA), leaf area index (LAI), days to heading (DH), days to maturity (DM), grain-filling duration (GFD), plant height (PH), spike length (SL), number of spikes (NS), number of spikelets (NSS), number of kernels (NKS), thousand-kernel weight (HKW), and grain yield (GY).

**Table 3.** Eigenvectors and percentage of accumulated contribution of principal components.

| | PCI1 | PCI2 | PCI3 | PCI4 | PCI5 | PCI6 | PCI7 |
|---|---|---|---|---|---|---|---|
| Eigenvalue | 9.221 | 4.157 | 2.829 | 2.053 | 1.352 | 1.138 | 1.114 |
| Variability (%) | 35.465 | 15.989 | 10.883 | 7.896 | 5.199 | 4.378 | 4.283 |
| Cumulative % | 35.465 | 51.454 | 62.336 | 70.232 | 75.431 | 79.809 | 84.092 |
| Eigenvectors: | | | | | | | |
| CT | −0.210 | 0.200 | 0.162 | 0.158 | 0.056 | 0.012 | −0.305 |
| LWC | 0.138 | 0.171 | 0.217 | 0.355 | 0.177 | 0.101 | 0.237 |
| RWC | 0.310 | 0.060 | 0.001 | 0.016 | 0.095 | −0.169 | −0.070 |
| Pn | 0.128 | −0.195 | −0.320 | 0.274 | −0.038 | 0.148 | 0.114 |
| Gs | −0.040 | 0.371 | −0.235 | −0.111 | −0.277 | −0.125 | 0.083 |
| Ci | −0.028 | 0.427 | −0.151 | 0.023 | 0.214 | 0.075 | −0.090 |
| E | −0.012 | 0.390 | −0.225 | −0.179 | −0.160 | −0.187 | 0.134 |
| Ls | 0.248 | −0.040 | 0.165 | −0.102 | 0.108 | −0.071 | 0.033 |
| POD | 0.102 | 0.070 | 0.085 | −0.314 | −0.391 | 0.517 | 0.132 |
| PPO | 0.181 | 0.086 | −0.236 | −0.327 | 0.024 | −0.122 | 0.324 |
| CAT | 0.225 | −0.070 | −0.256 | 0.294 | −0.089 | 0.167 | 0.014 |
| GLN | 0.246 | 0.053 | −0.019 | −0.311 | 0.174 | 0.150 | −0.318 |
| FLA | 0.233 | 0.175 | 0.131 | 0.065 | 0.065 | 0.171 | 0.278 |
| GLA | 0.145 | 0.098 | −0.169 | −0.001 | 0.559 | 0.371 | 0.143 |
| LAI | 0.193 | −0.081 | 0.290 | −0.096 | −0.244 | 0.151 | 0.009 |
| DH | 0.239 | 0.061 | 0.145 | −0.136 | 0.028 | −0.272 | −0.352 |
| MD | 0.288 | 0.059 | 0.131 | 0.065 | 0.093 | −0.079 | −0.138 |
| GFD | 0.186 | −0.192 | −0.261 | 0.168 | 0.031 | −0.194 | −0.138 |
| NS | 0.233 | 0.116 | 0.077 | −0.011 | −0.084 | 0.157 | −0.270 |
| PH | 0.276 | −0.044 | −0.133 | −0.127 | 0.082 | −0.119 | −0.034 |
| SL | 0.225 | 0.155 | 0.171 | 0.044 | −0.160 | 0.091 | −0.111 |
| NSS | 0.055 | 0.010 | 0.392 | −0.039 | 0.125 | −0.273 | 0.442 |
| NKS | 0.056 | 0.228 | 0.158 | 0.438 | −0.266 | 0.035 | −0.026 |
| HW | 0.257 | 0.100 | −0.035 | 0.168 | −0.155 | −0.337 | 0.149 |
| GY | 0.235 | −0.142 | −0.237 | 0.108 | −0.254 | 0.033 | −0.023 |

Canopy temperature (CT), leaf water content (LWC), relative water content (RWC), photosynthesis rate (Pn), stomatal conductance (Gs), intracellular $CO_2$ concentration (Ci), transpiration rate (E), stomatal limitation value (Ls), peroxidase (POD), polyphenol oxidase (PPO), catalase (CAT), green leaf number (GLN), flag leaf area (FLA), and green leaf area (GLA), leaf area index (LAI), days to heading (DH), days to maturity (DM), grain-filling duration (GFD), plant height (PH), spike length (SL), number of spikes (NS), number of spikelets (NSS), number of kernels (NKS), thousand-kernel weight (HKW), and grain yield (GY).

**Table 4.** The values of comprehensive index (PCIi), index weight, P μ($x_i$), D and regression D′ for each wheat genotype.

| Genotypes | The Values of Comprehensive Index (PCI$_i$) | | | | | | | Membership Function Value | | | | | | | D |
| | PCI$_1$ | PCI$_2$ | PCI$_3$ | PCI$_4$ | PCI$_5$ | PCI$_6$ | PCI$_7$ | P μ($x_1$) | P μ($x_2$) | P μ($x_3$) | P μ($x_4$) | P μ($x_5$) | P μ($x_6$) | P μ($x_7$) | Value |
|---|---|---|---|---|---|---|---|---|---|---|---|---|---|---|---|
| DHL12 | −0.449 | 1.653 | 0.442 | 1.106 | 0.901 | 0.023 | −0.370 | 0.508 | 0.629 | 0.694 | 0.708 | 0.754 | 0.318 | 0.381 | 0.510 |
| DHL02 | −0.474 | 4.236 | 0.920 | 2.481 | −1.287 | −0.491 | 1.226 | 0.506 | 1.000 | 0.780 | 1.000 | 0.256 | 0.195 | 0.766 | 0.638 |
| DHL25 | −0.100 | −1.192 | 0.085 | 0.607 | 0.077 | −0.143 | 2.194 | 0.538 | 0.220 | 0.630 | 0.602 | 0.567 | 0.278 | 1.000 | 0.458 |
| DHL07 | −0.455 | 0.579 | −0.132 | −0.576 | −0.699 | −0.814 | 1.068 | 0.508 | 0.475 | 0.591 | 0.350 | 0.390 | 0.117 | 0.728 | 0.451 |
| DHL26 | −4.260 | −2.042 | −1.307 | −2.225 | −0.509 | −0.405 | 0.760 | 0.178 | 0.098 | 0.379 | 0.000 | 0.433 | 0.215 | 0.654 | 0.176 |
| Gemmeiza-9 | −1.151 | −0.275 | 0.927 | 1.222 | 1.670 | 0.183 | 0.635 | 0.447 | 0.352 | 0.782 | 0.733 | 0.930 | 0.356 | 0.623 | 0.458 |
| DHL11 | −3.398 | 0.644 | 2.012 | 0.148 | −1.776 | −0.577 | −1.813 | 0.253 | 0.484 | 0.977 | 0.504 | 0.145 | 0.174 | 0.032 | 0.374 |
| KSU106 | −3.055 | −2.727 | 0.150 | 1.117 | 1.979 | 2.086 | −0.195 | 0.282 | 0.000 | 0.642 | 0.710 | 1.000 | 0.811 | 0.423 | 0.293 |
| Gemmeiza-12 | −3.858 | 1.622 | 2.139 | −1.332 | 1.873 | −0.452 | −0.286 | 0.213 | 0.625 | 1.000 | 0.190 | 0.976 | 0.204 | 0.401 | 0.377 |
| DHL01 | −6.311 | −0.143 | −2.197 | 1.552 | −0.591 | −0.665 | −0.435 | 0.000 | 0.371 | 0.219 | 0.803 | 0.415 | 0.153 | 0.365 | 0.193 |
| DHL14 | −0.906 | −0.395 | −0.419 | −1.227 | −2.410 | 2.878 | 0.532 | 0.468 | 0.335 | 0.539 | 0.212 | 0.000 | 1.000 | 0.598 | 0.381 |
| DHL29 | 2.112 | 2.032 | −2.439 | −1.936 | 1.441 | −0.849 | 0.989 | 0.730 | 0.683 | 0.175 | 0.062 | 0.877 | 0.109 | 0.709 | 0.503 |
| DHL15 | 2.674 | 0.093 | −3.411 | 0.491 | −0.425 | −0.787 | −0.282 | 0.779 | 0.405 | 0.000 | 0.577 | 0.452 | 0.124 | 0.402 | 0.480 |
| DHL06 | 1.781 | −1.032 | −2.367 | −0.815 | 0.173 | 0.159 | −1.258 | 0.701 | 0.244 | 0.188 | 0.300 | 0.589 | 0.350 | 0.166 | 0.404 |
| Misr1 | 1.812 | −2.579 | 1.851 | −0.721 | −0.101 | −0.588 | −0.118 | 0.704 | 0.021 | 0.948 | 0.320 | 0.526 | 0.171 | 0.441 | 0.476 |
| DHL05 | 2.589 | 2.069 | −1.560 | 1.179 | 0.565 | 1.023 | −1.944 | 0.771 | 0.689 | 0.333 | 0.723 | 0.678 | 0.557 | 0.000 | 0.569 |
| DHL23 | 3.730 | 3.361 | 1.646 | −1.906 | 0.079 | 1.464 | 0.305 | 0.870 | 0.874 | 0.911 | 0.068 | 0.567 | 0.662 | 0.544 | 0.686 |
| Sakha-93 | 1.116 | −0.655 | 1.634 | −1.723 | −0.290 | −1.305 | −1.433 | 0.644 | 0.298 | 0.909 | 0.107 | 0.483 | 0.000 | 0.124 | 0.462 |
| Pavone-76 | 3.377 | −2.693 | 1.080 | 0.542 | −0.323 | −0.456 | 0.345 | 0.840 | 0.005 | 0.809 | 0.588 | 0.476 | 0.203 | 0.553 | 0.543 |
| DHL08 | 5.228 | −2.556 | 0.946 | 2.017 | −0.346 | −0.282 | 0.080 | 1.000 | 0.025 | 0.785 | 0.901 | 0.470 | 0.245 | 0.489 | 0.638 |

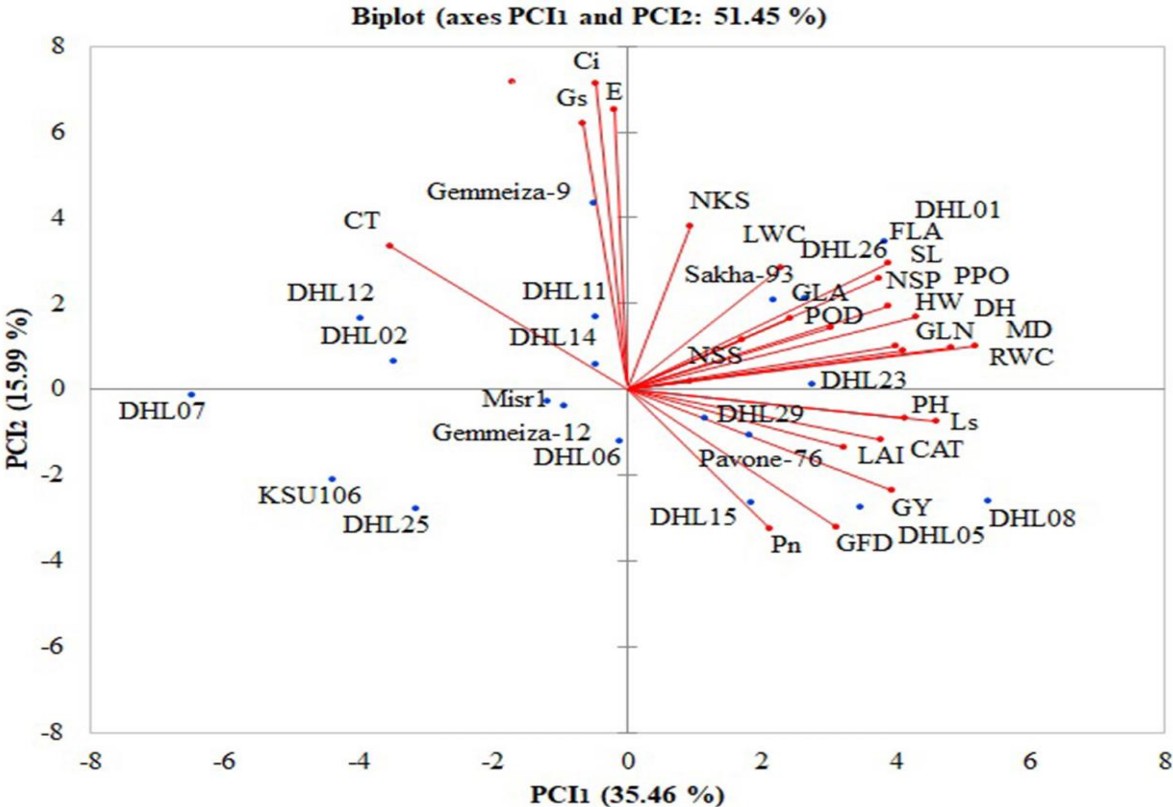

**Figure 1.** Biplot for the first two principle components in the principle component analysis of 20 wheat genotypes in two seasons (S1 and S2) for 25 heat tolerance indices.

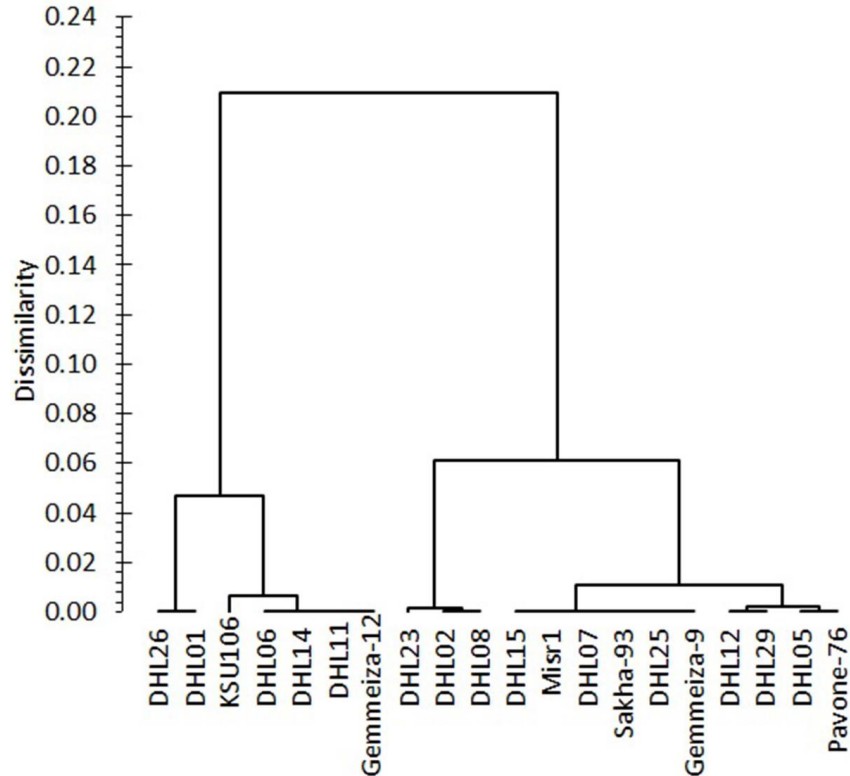

**Figure 2.** Dendrogram showing clustering of 20 wheat genotypes based on the Euclidean distance of the (D) value.

Cluster I was classified as highly tolerant (HT, with the highest value of D ≥ 0.638), consisting of three genotypes (DHL02, DHL23, DHL08); cluster II was classified as tolerant (T, with values of 0.638 > D ≥ 0.503), consisting of four genotypes (DHL12, DHL29, DHL05, and Pavone-76), Cluster III was classified as moderately tolerant (I, with values of 0.503 > D ≥ 0.458), consisting of six genotypes (DHL25, DHL07, Gemmeiza-9, DHL15, Misr1, and Sakha-93), Cluster IV was classified as sensitive (S, with values of 0.458 > D ≥ 0.293), consisting of five genotypes (DHL11, KSU106, Gemmeiza-12, DHL14, and DHL06). The genotypes DHL26 and DHL01 had lower D values (D < 0.293) and were placed into Cluster VI and designated as a highly sensitive (HS) group (Figure 2).

### 3.2.2. Identification of Indices Related to Yield Tolerance Index

We investigated the relationship between the heat tolerance indices (HTIs) of each index with the GY index. Seven HTIs (MD, GFD, PH, RWC, SL, HW, and Pn) showed a significant positive correlation, but the CT index was a significant negative correlation (Figure 3). The relationships between all indices were analyzed using SMLR in each genotype, as independent indices, in order to understand the best-measured and heat tolerance-related indices and their contribution to GY index performance as a dependent index (Table 5). The results of SMLR showed that indices GFD, SL, and CT only from the eight were directly relevant to the GY index ($R^2$ of the SMLR model was 0.739, $p < 0.0001$), and their contribution rates were 0.529, 0.074, and 0.136, respectively (Table 5). The three indices of the GY index variation are partitioned into direct effects of each index and multiple indirect effects with the other indices using path coefficient analysis.

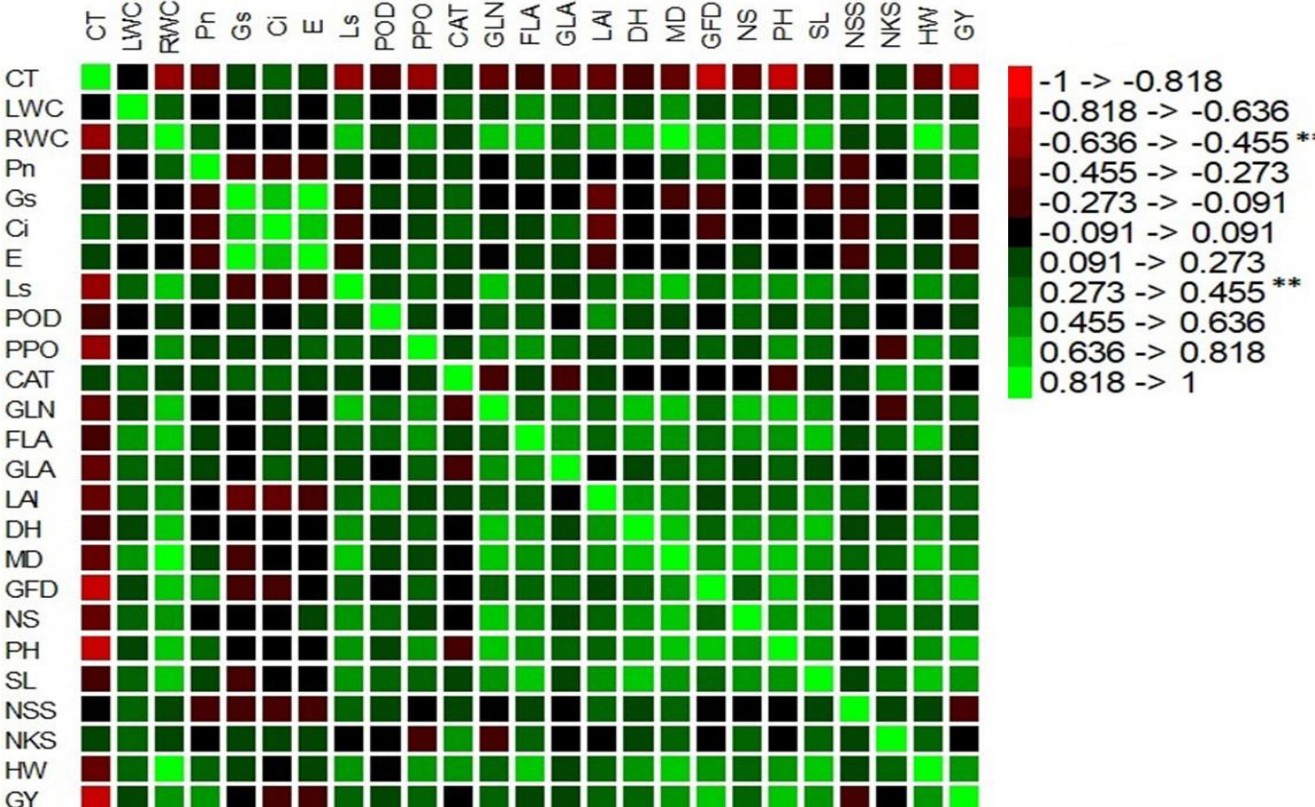

**Figure 3.** Correlation matrix among 25 indices measured. ** indicate significance at $p < 0.01$.

**Table 5.** Stepwise regression and path coefficient analyses for grain yield (dependent index) with three yield-related traits (independent indices) for combined data across the two seasons.

| Source | Stepwise Regression | | | | Path Coefficient | | | $R^2$ |
| | | | | | Partitioning the Correlation | | | |
| | Regression Coefficient | *p*-Value | $R^2$ Par. | $R^2$ Com. | Direct Effect | Indirect Effect | Correlation Value | Direct Effect |
|---|---|---|---|---|---|---|---|---|
| Intercept | 0.670 | 0.033 | | | | | | |
| GFD | 0.504 | **0.007** | 0.529 | 0.529 | 0.460 | 0.345 | **0.805** | 0.211 |
| SL | 0.334 | **0.049** | 0.074 | 0.603 | 0.276 | 0.222 | **0.498** | 0.076 |
| CT | −0.466 | **0.009** | 0.136 | 0.739 | −0.437 | −0.241 | **−0.677** | 0.191 |
| Indirect effect | | | | | | | | 0.261 |
| Total $R^2$ | | | | 0.739 | | | | 0.739 |
| Residual | | | | 0.511 | | | | 0.511 |

Coefficient partial determination ($R^2$ Par.), cumulative coefficient determination ($R^2$ Com.), Values in bold indicate at significance.

This contribution had a direct influence value of 478 (GFD, alone possessed 0.211 of them) and an indirect influence value of 0.261. The $R^2$ value was (0.739), which is the same as the SMLR model, with a noise value of 0.511, suggesting that these three indices could be used as important selection criteria to determine the extent of the tolerance of wheat genotypes for heat. According to Equation of the model (GY, Table 5), GY = 0.670 + 0.504 × GFD + 0.334 × SL − 0.466 × CT, the predicted regression GY value, error value, and relative error value ranged from 0.663 to 0.972, from −0.142 to 0.081, from −0.261 to 0.105, respectively. Evaluation accuracy (%) ranged from 73.917 to 98.570, with an average value of 94.12 (Table 6).

**Table 6.** The stepwise linear regression analysis was applied to establish optimal regression equation (GY) for predictions, residuals, and evaluation accuracy (%).

| Genotypes | Dependent Indices | | | GY | Regression "GY" Value | Predicted Error Value | Relative Error Value | Evaluation Accuracy (%) |
| | GFD | SL | CT | | | | | |
|---|---|---|---|---|---|---|---|---|
| DHL12 | 0.694 | 0.845 | 1.315 | 0.769 | 0.689 | 0.081 | 0.105 | 89.49 |
| DHL02 | 0.695 | 0.896 | 1.286 | 0.736 | 0.720 | 0.016 | 0.021 | 97.86 |
| DHL25 | 0.798 | 0.907 | 1.078 | 0.840 | 0.873 | −0.032 | −0.038 | 96.16 |
| DHL07 | 0.783 | 0.751 | 1.007 | 0.811 | 0.846 | −0.035 | −0.043 | 95.72 |
| DHL26 | 0.694 | 0.709 | 1.171 | 0.773 | 0.711 | 0.062 | 0.080 | 92.03 |
| Gemmeiza-9 | 0.758 | 0.752 | 1.155 | 0.775 | 0.764 | 0.011 | 0.014 | 98.57 |
| DHL11 | 0.606 | 0.870 | 1.293 | 0.648 | 0.663 | −0.015 | −0.023 | 97.73 |
| KSU106 | 0.667 | 0.707 | 1.199 | 0.662 | 0.683 | −0.021 | −0.032 | 96.85 |
| Gemmeiza-12 | 0.709 | 0.794 | 1.300 | 0.545 | 0.687 | −0.142 | −0.261 | 73.92 |
| DHL01 | 0.725 | 0.640 | 1.250 | 0.729 | 0.666 | 0.063 | 0.086 | 91.39 |
| DHL14 | 0.751 | 0.863 | 1.137 | 0.846 | 0.807 | 0.039 | 0.046 | 95.36 |
| DHL29 | 0.783 | 0.805 | 1.083 | 0.736 | 0.829 | −0.093 | −0.126 | 87.37 |
| DHL15 | 0.975 | 0.859 | 1.086 | 0.929 | 0.942 | −0.013 | −0.014 | 98.57 |
| DHL06 | 0.817 | 0.851 | 1.029 | 0.919 | 0.886 | 0.033 | 0.036 | 96.41 |
| Misr1 | 0.783 | 0.850 | 1.082 | 0.820 | 0.844 | −0.024 | −0.029 | 97.08 |
| DHL05 | 0.852 | 0.929 | 1.198 | 0.920 | 0.851 | 0.069 | 0.075 | 92.53 |
| DHL23 | 0.715 | 0.857 | 1.099 | 0.777 | 0.805 | −0.028 | −0.036 | 96.40 |
| Sakha-93 | 0.757 | 0.863 | 1.185 | 0.814 | 0.787 | 0.027 | 0.033 | 96.69 |
| Pavone-76 | 0.898 | 0.903 | 1.090 | 0.869 | 0.916 | −0.047 | −0.054 | 94.55 |
| DHL08 | 0.929 | 0.929 | 1.022 | 0.951 | 0.972 | −0.022 | −0.023 | 97.73 |
| Average | | | | | | | | 94.12 |

### 3.2.3. Clustering and Genetic Relationships between the Genotypes for Heat Tolerance

Based on SMLR analysis, we decided to use the tolerance index of the four indices (GFD, CAT, SL, and GY) for cluster analysis for heat tolerance of 20 wheat genotypes using genetics dissimilarity matrix (Euclidean distance and Ward's method of agglomeration). Cluster analysis showed five major groups based on the heat tolerance range of wheat genotypes with a dissimilarity coefficient of 2.130. Cluster I, classified as highly tolerant (HT), consists of four genotypes (DHL05, DHL23, Pavone-76, and DHL08); cluster II or tolerant (T) with six genotypes (DHL12, DHL25, DHL29, DHL06, Misr1, and Sakha-93), cluster III of moderately tolerant (I) two genotypes (DHL07 and DHL15), cluster IV or sensitive (S), of four genotypes (DHL02, DHL11, Gemmeiza-12, and DHL14). Cluster VI was classified as highly sensitive (HS), consisting of four genotypes (DHL26, Gemmeiza-9, KSU106, and DHL01) (Figure 4).

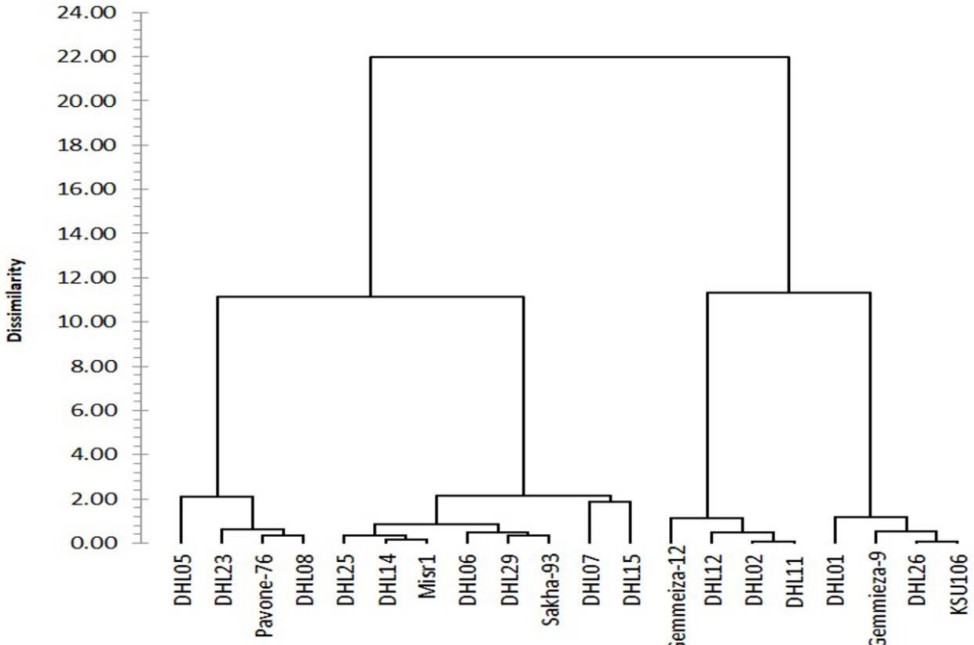

**Figure 4.** Dendrogram showing clustering of 20 wheat genotypes based on the Euclidean distance of four tolerance indices: GFD, CAT, SL, and GY.

The clustering result was substantially aligned with our phenotypic results. The classification of heat tolerance index for both PCA and clustering were significantly correlated (r = 0.502, $p < 0.0001$), building upon the Mantel test. Phenotypic variation between the five heat groups indicated significantly different performances for four indices (Figure 5). The mean comparison for each index showed significant differences in HT with S and HS groups for four indices, except for group S, which was insignificant for the SL index. There were insignificant differences between the T and I groups, except for group I, which was significant for the SL index. Group T vs. I exhibited insignificant differences in four indices. Group S vs. HS exhibited insignificant differences in GFD and GY indices and SL and CT indices (Figure 5).

### 3.2.4. Differentiation of Heat Groups by Discriminant Function Analysis

Multicollinearity statistics were acceptable, ranging from 0.278 to 0.696 for tolerance; and 1.436 to 3.598 for VIF (Table 7). Unidimensional test of equality of the means of the five groups by four indices (GFD, SL, CT, and GY) was significant with a low Lambda level, with values ranging from 0.216 ($p < 0.0001$) to 0.404 ($p < 0.006$). This indicates the possibility of predicting the performance of group members based on cluster analysis. The discriminant functions (two-dimensional) of five groups and four indices were closely related to the

prediction of group members into the heat groupings for the 20 genotypes used in this study (Figure 4). The first and second canonical discriminant functions (Can) explained 54.325% and 43.385% (cumulative, 97,710), respectively, of the overall phenotypic variations (eigenvalues > 1) (Table 8). Bartlett's statistic was significant for the first (54.926, *p* = 0.000) and second (27.670, *p* = 0.001) canonical. Canonical correlations were highly significant at 0.921 (Can1) and 0.903 (Can2). Loading the four indices to canonical discriminant functions (DF-Can1) showed that SL and CT were positively correlated, and GFD and GY were negatively correlated, reflecting that DF-Can1 discriminated between genotypes based on SL and CT. DF-Can2 was inextricably linked to GFD, SL, and GY but negatively correlated to CT, reflecting that DF-Can2 discriminates the genotypes based on GFD, SL, and GY.

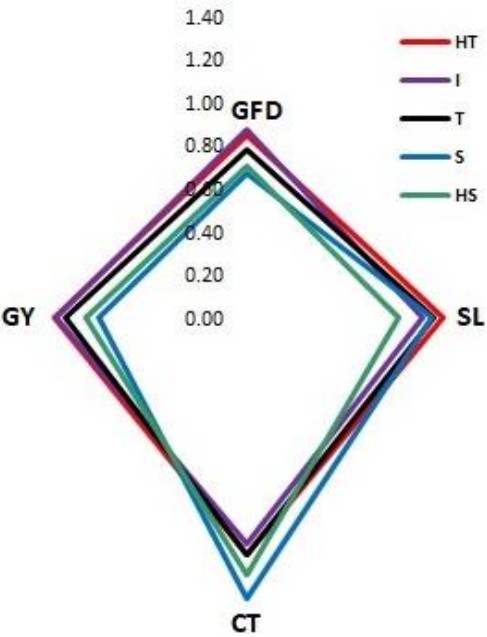

**Figure 5.** Radar charts comparing three indices of the five groups. Data was analyzed using least squares (LS). Grain-filling duration (GFD), canopy temperature (CT), spike length (SL), and grain yield (GY).

The maximum distinction between group means I and S (3.681 vs. −2.940) was observed in DF-Can1, and between group means HT and HS (2.056 vs. −3.080) was observed in DF-Can2. S group with a positive DF-Can1 value was heat sensitive; conversely, four groups (HS, I, T, and HT) with DF-Can1 negative values. In the plot of heat groups with DF-Can1 and DF-Can2, group I was placed halfway between HS and HT (Figure 6). Group S had a positive value of DF-Can1 (3.681), indicating that it had high mean values for SL and CT indices and a negative DF-Can2 (−0.949) value due to low CT. The HT, T, and I groups were against the S group, which had a negative value for DF-Can1 (−0.017, −0.357, and −2.940) and a positive value for DF-Can2 (2.056, 1.088, and 0.682), respectively. The HS group had negative values for the two DF-Can (−1.659 and −3.080), respectively (Table 8). After finding the DF-Can, the classification of the five groups was verified, which showed that prior and posterior classification was identical in eighteen genotypes (% correct = 90%) and different in the two genotypes (DHL25 and DHL23). The values of membership probabilities (>0.5) indicated compatibility between prior and posterior classification (Table 9). The membership probabilities values = 1 in five genotypes (DHL02, DHL26, DHL11, Gemmeiza-12, and DHL01) and when it is less than 0.5 to be transferred to the appropriate classification. Cross-validation showed that prior and posterior classification was identical in thirteen genotypes (% correct = 65%) and different in the seven genotypes (DHL25, DHL07, DHL15, DHL06, DHL05, DHL23, and DHL08) (Table 9).

**Table 7.** Multicollinearity statistics and unidimensional test of equality of the means of the classes of heat group to canonical discriminant function.

| Statistic | GFD | SL | CT | GY |
|---|---|---|---|---|
| **Multicollinearity statistics:** | | | | |
| Tolerance | 0.321 | 0.696 | 0.465 | 0.278 |
| VIF | 3.114 | 1.436 | 2.151 | 3.598 |
| **Unidimensional test of equality of the means of the classes:** | | | | |
| Lambda | 0.377 | 0.220 | 0.216 | 0.404 |
| F | 6.195 | 13.319 | 13.584 | 5.536 |
| DF1 | 4 | 4 | 4 | 4 |
| DF2 | 15 | 15 | 15 | 15 |
| *p*-value | 0.004 | <0.0001 | <0.0001 | 0.006 |

**Table 8.** Total canonical structure of eigenvalue, canonical discriminant function, bartlett's statistic, and class means of heat group to canonical discriminant function.

| Parameters | Can1 | Can2 | Can3 |
|---|---|---|---|
| Eigenvalue | 5.551 | 4.433 | 0.200 |
| Discrimination (%) | 54.325 | 43.385 | 1.959 |
| Cumulative % | 54.325 | 97.710 | 99.669 |
| Bartlett's statistic | 54.926 | 27.670 | 3.128 |
| *p*-value | 0.000 | 0.001 | 0.537 |
| Canonical correlations | 0.921 | 0.903 | 0.408 |
| **Variables/Factors correlations:** | | | |
| GFD | −0.497 | 0.668 | 0.543 |
| SL | 0.425 | 0.875 | 0.062 |
| CT | 0.732 | −0.629 | 0.181 |
| GY | −0.505 | 0.673 | 0.083 |
| **Heat Group** | | | |
| HS | −1.659 | −3.080 | −0.044 |
| HT | −0.017 | 2.056 | 0.312 |
| I | −2.940 | 0.682 | 0.667 |
| S | 3.681 | −0.949 | 0.173 |
| T | −0.357 | 1.088 | −0.516 |

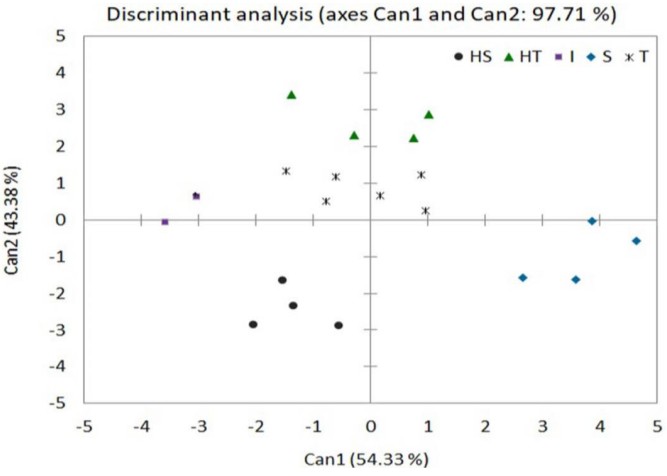

**Figure 6.** Distribution of 20 wheat genotypes by discriminant analysis of grain-filling duration, canopy temperature, spike length, and grain yield indices. Highly tolerant (HT), tolerant (T), intermediate (I), sensitive (S), and highly sensitive (HS).

**Table 9.** Posterior probability of membership in heat groupings by linear discriminant analysis.

| Genotypes | | | Classification | | | | | | Cross-Validation | | | | | |
|---|---|---|---|---|---|---|---|---|---|---|---|---|---|---|
| | | | Membership Probabilities | | | | | | | Membership Probabilities | | | | |
| | Prior | Posterior | Pr(HS) | Pr(HT) | Pr(I) | Pr(S) | Pr(T) | Posterior | HS | HT | I | S | T |
| DHL12 | S | S | 0.000 | 0.000 | 0.000 | 0.999 | 0.001 | S | 0.001 | 0.004 | 0.000 | 0.972 | 0.023 |
| DHL02 | S | S | 0.000 | 0.000 | 0.000 | 1.000 | 0.000 | S | 0.000 | 0.000 | 0.000 | 0.999 | 0.001 |
| DHL25 | T | **HT** | 0.000 | 0.520 | 0.000 | 0.000 | 0.480 | **HT** | 0.000 | 0.703 | 0.000 | 0.000 | 0.297 |
| DHL07 | I | I | 0.008 | 0.001 | 0.955 | 0.000 | 0.036 | **HS** | 0.637 | 0.000 | 0.000 | 0.000 | 0.363 |
| DHL26 | HS | HS | 1.000 | 0.000 | 0.000 | 0.000 | 0.000 | HS | 0.999 | 0.000 | 0.001 | 0.000 | 0.001 |
| Gemmeiza-9 | HS | HS | 0.928 | 0.001 | 0.029 | 0.000 | 0.042 | HS | 0.865 | 0.001 | 0.062 | 0.000 | 0.072 |
| DHL11 | S | S | 0.000 | 0.000 | 0.000 | 1.000 | 0.000 | S | 0.000 | 0.000 | 0.000 | 1.000 | 0.000 |
| KSU106 | HS | HS | 0.999 | 0.000 | 0.000 | 0.000 | 0.001 | HS | 0.993 | 0.000 | 0.000 | 0.000 | 0.006 |
| Gemmeiza-12 | S | S | 0.000 | 0.000 | 0.000 | 1.000 | 0.000 | S | 0.001 | 0.000 | 0.000 | 0.999 | 0.000 |
| DHL01 | HS | HS | 1.000 | 0.000 | 0.000 | 0.000 | 0.000 | HS | 1.000 | 0.000 | 0.000 | 0.000 | 0.000 |
| DHL14 | T | T | 0.000 | 0.169 | 0.001 | 0.000 | 0.830 | T | 0.000 | 0.215 | 0.001 | 0.000 | 0.783 |
| DHL29 | T | T | 0.001 | 0.112 | 0.043 | 0.000 | 0.844 | T | 0.003 | 0.177 | 0.129 | 0.000 | 0.691 |
| DHL15 | I | I | 0.000 | 0.105 | 0.845 | 0.000 | 0.050 | **HT** | 0.000 | 0.894 | 0.000 | 0.000 | 0.106 |
| DHL06 | T | T | 0.000 | 0.114 | 0.226 | 0.000 | 0.660 | **I** | 0.000 | 0.100 | 0.742 | 0.000 | 0.158 |
| Misr1 | T | T | 0.000 | 0.188 | 0.008 | 0.000 | 0.804 | T | 0.000 | 0.204 | 0.010 | 0.000 | 0.787 |
| DHL05 | HT | HT | 0.000 | 0.755 | 0.000 | 0.003 | 0.242 | **T** | 0.000 | 0.273 | 0.000 | 0.035 | 0.692 |
| DHL23 | HT | **T** | 0.000 | 0.133 | 0.000 | 0.000 | 0.867 | **T** | 0.000 | 0.003 | 0.000 | 0.001 | 0.996 |
| Sakha-93 | T | T | 0.000 | 0.231 | 0.000 | 0.020 | 0.749 | T | 0.001 | 0.418 | 0.000 | 0.025 | 0.556 |
| Pavone-76 | HT | HT | 0.000 | 0.766 | 0.005 | 0.000 | 0.230 | HT | 0.000 | 0.671 | 0.009 | 0.000 | 0.320 |
| DHL08 | HT | HT | 0.000 | 0.813 | 0.018 | 0.000 | 0.170 | **T** | 0.000 | 0.395 | 0.181 | 0.000 | 0.424 |

Bold letters indicate misclassified wheat genotypes.

## 4. Discussion

Plant responses to stress are very complicated; it incorporates a set of temporal scales, from in just a few seconds to largely noticed evolutionary processes. We understand at least three clear time scales of plant response to stress, i.e., response to stress, acclimation, and adaptation. A plant's response to stress varies with the developmental stage and genotype used, and the plant's ability to survive under stress conditions is associated with stress tolerance [45]. Developing heat-tolerant genotypes is among the main goals of plant breeders, given that the final grain yield is affected by many genetic and environmental factors. Plant breeders rely on explained attributes associated with GY as screening criteria, which may provide greater information in the selection process of more sustainable and heat-tolerant genotypes [46,47]. There is a crucial need for a comprehensive understanding of the indirect approach behavior based on incorporating multiple morpho-physiological plant traits associated with heat tolerance with yield and its components. This can assist in targeting major traits, which may help genetic gain for grain yield for countries with high ambient temperatures. The accurate selection of explained attributes is efficient if there are strong associations with GY and high-value heritability and genetic gain [45,48,49].

Based on one-way ANOVA for heat tolerance index in S1 and S2, there were highly significant differences ($p < 0.01$) for genotypes for all indices, with a significant variation between large and small values for the majority of measured indices, reflecting the efficiency of these indices on the show genetic diversity between genotypes used (Tables 1 and 2). The interactions in combined ANOVA between genotypes and seasons were significant, with nineteen indices out of twenty-six, suggesting that the genotypes' heat tolerance index differed from season to season (Table 1). Test of the homogeneity between two seasons showed insignificant for 22 indices, suggesting that the combined data can be shown without exposing seasons data. Plant breeder depends primarily on the trait's genetic stability (heritability, genetic gain and GCV). The 17 studied indices showed high heritability ($h^2 > 60\%$), genetic gain (nine of them were high >20% and/or eight of them were moderate >10%), and GCV and PCV were very close. This indicates that the genotypic variations primarily arise from the genetic control indices. These indices, therefore, can reliably be used as selection criteria for the evaluation of heat tolerance [28,33,48], notably if the measurement method is quick, easy, and low-budget [34]. The values for coefficients of experimental variation (CVe) were small, ranging from 3.94% to 5.43 (Table 2). These results show small variability within experimental units, and the number of replicates used was appropriate [50,51].

The studied wheat genotypes exhibited wide variation in heat tolerance. The stress tolerance index, typically used as a criterion to evaluate tolerance, has been used for screening genotypes in multiple recent literatures [29,35,45,52]. They have relied on quite fewer genotypes compared to our study. Although the number of applied genotypes could sound insufficient, it gave rise to reliable selection criteria and accurate results when multivariate analysis techniques and multidimensional methods were conducted for discriminating their heat tolerance [29,52–54]. In this study, we conducted a comprehensive assessment of wheat heat tolerance by many multivariate analyses with a view to achieving a more accurate classification. The angle between the vectors of indices was acute (less than 90°) for the GY index with most indices, which indicates a positive correlation of these traits with GY. In contrast, the angle between the vector of the four indices (CT, Gs, Ci, and E) was for GY higher than 90°, which indicated their negative correlation with GY (Figure 1). By PCA, the 25 indices were transformed into seven comprehensive indices (eigenvalue > 1) (Table 3), and the weight factor for each comprehensive index was determined (Table 3). Then, for each wheat genotype, the comprehensive evaluation value D to identify heat-tolerant capacity was calculated by the combination membership function analysis (Table 4). A high D value indicated higher heat tolerance capacity. Thus, the D value and the results of hierarchical cluster analysis were used to evaluate the heat tolerance of the 20 tested genotypes (Figure 2). The heat tolerance of wheat was determined, and studied genotypes were divided into five groups—three genotypes HT

(DHL02, DHL23, DHL08), four genotypes T (DHL12, DHL29, DHL05, and Pavone-76), six genotypes I (DHL25, DHL07, Gemmeiza-9, DHL15, Misr1, and Sakha-93), four genotypes S (DHL11, KSU106, Gemmeiza-12, DHL14 and DHL06), and two genotypes HS (DHL26 and DHL01). This method was more confident than other traditional assessment metrics [52,55].

Heat tolerance is a comprehensive quantitative trait and a complex trait affected by many genetic and environmental factors and their interactions [56]. Thus, using only one index poorly reflects wheat genotypes' heat tolerance, and the significant correlations between indices indicate overlap with others to heat stress [1,35]. We used other statistical analyses to further the accuracy of our results, i.e., the use of 16 morph-physiological and agronomic indices (MD, GFD, FLA, SL, LWC, RWC, CT, Pn, Gs, Ci, E, Ls, POD, PPO, CAT, and GY) as stress indicators for screening heat-tolerant genotypes. Out of these, nine indices, i.e., DH, NS, PH, GLN, GLA, LAI, NSS, NKS, and HW, were excluded from other statistical analyses because of their low heritability level and/or genetic gain with wide variation between GCV and PCV. Moreover, the 16 single HTIs showed highly significant differences among the 20 tested genotypes (Table 1). The seven indices of MD, GFD, PH, RWC, SL, HW, and Pn were correlated significantly positively with the GY index, while the CT index had a negative correlation (Figure 3). SMLR and path coefficient are effective tools for understanding the relationships between independent and dependent variables [1,57]. They have also found that a simple correlation analysis with interactions between independent and dependent indices may not help successful breeding programs [58]. We analyzed the relationships between 16 indices using SMLR for each genotype as independent indices to understand the best-measured and heat tolerance-related indices and their contribution to GY index performance as a dependent index (Table 5). The SMLR results showed that indices GFD, SL and CT only from the eight were directly relevant to the GY index ($R^2$ of the SMLR model was 0.739, $p < 0.0001$), and their contribution rates were 0.529, 0.074, and 0.136, respectively (Table 5). We further conducted path analysis to separate the three indices (GFD, SL, and CT) for their direct and indirect impact. A direct effect of the correlation between the interpreted indices shows a direct connection among these indices and suggests their use in the selection process [1,59]. We found that the direct and indirect impacts were close for GDF and SL, but for CT, the direct impact was two-fold the indirect impact (Table 5).

The $R^2$ values were 0.478 and 0.261 for direct and indirect impacts, respectively, and most of the impact was directly related to the GFD index. So, we realized that GFD, SL, and CT are good indices for predicting yield index and dependable to appraise the genotypic heat tolerance for their valuable contribution to the predicting process (Table 5). Naturally, the different genotypes perform from one index to another, but at least it will relate to one index [1]. In the same genotype, some traits might show a positive correlation, whereas others may show a negative correlation. As a result, we noticed the equation of the model (GY, Table 5), GY = $0.670 + 0.504 \times GFD + 0.334 \times SL - 0.466 \times CT$, and which showed that the predicted regression GY value, error value, and relative error value ranged from 0.663 to 0.972, from $-0.142$ to 0.081, from $-0.261$ to 0.105, respectively, with genotypes evaluation accuracy (%) ranging from 73.917 to 98.570 with an average value of 94.12 (Table 6), apparently, according to Yu et al. [52]. Indices such as GFD, SL, and CT are valuable indicators of overall plant performance; they consider radiation use efficiency, plant competition, photosynthesis, and evapotranspiration rates for computing plant growth and development [17,33]. Increasing GFD (from anthesis to maturity) is a critical target in plant breeding under heat stress. Therefore, plant breeders seek to break the negative correlation between DH and GY by obtaining high-yielding and early flowering bread wheat genotypes under heat stress [1]. GFD is a robust index for identifying genotypes of heat tolerance, given the interdependencies with morpho-physiological and early traits under heat stress. It has a strong relationship with SL and helps photosynthesis through awns. In this vein, CT is a strong index because closely linked to photosynthetic capacity, chlorophyll content, and several traits connected with high heat in plants [60–62]. The genotypes with the capacity to reduce CT and gas exchange are highly desirable, because

of the efficiency indicator in gas exchange and transpiration as a leaf-cooling response under stress [63,64]. CT is a robust physiological index used in wheat breeding programs as a low-budget nondestructive tool for identifying heat-tolerant genotypes [60,61,65]. In wheat, genotypic variations for CT suggest the capacity of wheat genotypes also varies for transporting water cross-vascular system, regulating stomata aperture, metabolism, root biomass and depth, and source–sink balance [66]. Hence, the low CT trait should be the key focus of plant breeding programs for screening heat-tolerant wheat genotypes.

Based on the results of the SMLR analysis, we used the four tolerance indexes (GFD, CAT, SL, and GY) to create a cluster analysis for heat tolerance in the 20 wheat genotypes using a genetics dissimilarity matrix (Figure 4). The heat tolerance index for both PCA and clustering were significantly correlated (r = 0.502, $p < 0.0001$), building upon the Mantel test. Cluster analysis showed five major groups based on the tolerance range of wheat genotypes for heat. The HT, S, and HS groups represented four genotypes each, and T and I groups represented six and two genotypes, respectively. Six genotypes (DHL25, KSU106, Misr1, DHL05, Sakha-93, and Pavone-76) within groups showed deviation out to the nearest group compared to the classification of based comprehensive evaluation value D, but the genotypes (DHL12, DHL02, Gemmeiza-9, DHL14, and DHL06) showed deviation a lot further. Nine (DHL07, DHL26, DHL11, Gemmeiza-12, DHL01, DHL29, DHL15, DHL23, and DHL08) genotypes out of 20 were completely identical in the same categories obtained from comprehensive evaluation value D. Phenotypic variation between the five heat groups by one-way ANOVA indicated significantly different performances for the four indices (Figure 5). The group HT vs S in the SL index, HT vs T and I in GFD, CT, and GY indices, T vs I in the four indices, and S vs HS in GFD and GY indices had the same heat tolerance responses.

Many researchers have used cluster analysis for ranking the tolerant wheat genotypes based on their agro-physiological parameters [6,28,29,67,68]. However, cross-validation of the aggregation method was not employed to enhance classification reliability. In addition, identifying the differences between tolerance levels is unstable because heat tolerance is a complex inheritance and limited screening techniques [35,45,69]. So, FA was used to understand aggregation better and assess the extent of distinctions between heat categories to enhance classification reliability for heat tolerance. Fisher linear discriminant analysis (FLDA) is similar to MANOVA work; in the beginning, it computes the Mahalanobis distance of each genotype to a category and then uses this distance to classify a genotype into the category with the smallest generalized squared distance [68,70]. The homogeneity test for covariance matrices was significant ($0.043 < p < 0.0001$), and we conducted quadratic discriminant analysis, which resulted in a 0.00% error rate. This confirms that the classification of genotypes using clustering based on normalization values was a robust analysis. Lambda values were low ($0.006 < p < 0.0001$), which indicated the possibility of predicting group members based on the cluster analysis (Table 7). As indicated by DF, there were also strong contributions (cumulative, 97.71) and clear separations between the heat tolerance categories (Table 8). Our study suggested that DF-Can1 differentiates the genotypes based on SL and CT while DF-Can2 discriminates based on GFD, SL, and GY. The classification of the five groups was verified, which showed that prior and posterior classification was identical in eighteen genotypes (% correct = 90%) and different in the two genotypes (DHL25 and DHL23) (the values of membership probabilities > 0.5) (Table 9). Cross-validation showed that prior and posterior classification was identical in thirteen genotypes (% correct = 65%) and different in the seven genotypes (DHL25, DHL07, DHL15, DHL06, DHL05, DHL23, and DHL08). Therefore, DF could serve as a useful statistical tool for identifying genetic resources of heat tolerance using accurate and credible selection criteria [29,35,68]. Since heat-tolerant trait is very complicated and affected by many genetic and environmental factors and their interactions, an overall understanding of the genetic basis and plant responses to this stress and their interaction with the environment is needed [6,71–73]. We will be using recently discovered statistical methods in a future study with more environmental indices such as the multi-trait genotype-ideotype distance

index (MGIDI), the weighted average of absolute scores (WAASB) index, and a superiority index that allows weighting between mean performance and stability (WAASBY) to help breeders make appropriate decisions for selecting more stable genotypes and commended in multi-environment trials [72–74].

### 5. Conclusions

In this study, we studied genotypic × environmental interactions in wheat using 25 indices with homogeneity of error variance in 22 indices. We found that 16 indices combined high heritability and genetic gain, which can be used as indicators for screening heat-tolerant genotypes. The classification D value and SMLR distances were significantly correlated based on the Mantel test, with a perfect match in nine genotypes. The SMLR-based classification of studied wheat genotypes into five distinct groups was verified through discriminant analysis, which showed that prior and posterior classification was identical in eighteen genotypes and different in the two genotypes. Cross-validation showed that prior and posterior classification was identical in thirteen genotypes and different in the seven genotypes. The tolerated new wheat lines (DHL25, DHL05, DHL23, and DHL08) and cultivar Pavone-76 could be recommended as a promising genetic source for heat-tolerant breeding programs.

**Supplementary Materials:** The following supporting information can be downloaded at: https://www.mdpi.com/article/10.3390/agronomy13010154/s1, : title; Table S1: Monthly temperature data at the experimental location during the growing seasons. Table S2. Names and pedigree of the 20 bread wheat genotypes (6 cultivars and 14 doubled haploid lines (DHLs)) used in this study.

**Author Contributions:** Conceived and designed the experiments, I.A.-A.; performed the experiments, I.A.-A., A.I., M.S.; analyzed the data, I.A.-A., M S., A.A.-D.; morpho-physiological measurements, I.A.-A., M.S., A.I., M.A. and A.G.; edited the manuscript, I.A.-A. and N.U.; final approval of the version to be published, I.A.-A. All authors have read and agreed to the published version of the manuscript.

**Funding:** The authors extend their appreciation to the Deputyship for Research & Innovation, Ministry of Education in Saudi Arabia for funding this research work through the project no. (IFKSURG-2-4).

**Data Availability Statement:** All data is contained within the article or Supplementary Material.

**Acknowledgments:** The authors extend their appreciation to the Deputyship for Research & Innovation, Ministry of Education in Saudi Arabia for funding this research work through the project no. (IFKSURG-2-4).

**Conflicts of Interest:** The authors declare no conflict of interest.

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
