# Peer review of "Agro-Physiological Indices and Multidimensional Analyses for Detecting Heat Tolerance in Wheat Genotypes"

_agronomy, doi:10.3390/agronomy13010154_

Round 1

Reviewer 1 Report

A valuable practical study. However, there are a number of points that could be improved.

Firstly, the abstract is very long and should be presented in more concentrated form. In part of Materials and methods part 2.1 is very unclear. Rows 150-151 present information about fertilizers applied but presented amounts (regarding to dimentions) are unbelievable, please check it. Row 152 presents information about weather conditions but such form is null and void as there is no real information available, an additional file with meteorological information could be provided. Tables 2, 3, 4, 6, 9 are very broad and detailed but analysing their information is tricky. Could you convert come of them to figures or convert to tables with more concentrated information. Many traits have been analysed, but it should also be valuable to provide practical options on which traits can be practically applied to assess hundreds or thousands of genotypes.  I also suggest that the manuscript could be shortened by analysing the most important features and mentioning the less valuable ones briefly, or that some of the information could be moved to the supplementary files. 

Author Response

  • Dear Reviewer1

    Thank you very much for the valuable comments obtained. Kindly, find our response to the comments point by point below. Also, the changes have been made in the MS as track changes.  

  • A valuable practical study. However, there are a number of points that could be improved.

Response: Thank you for your comment. We greatly appreciate your critical and valuable observations as well as your constructive and helpful comments, which would show our manuscript in great shape.  We hope that we could address your comments /questions through the explanations and revisions made in the manuscript. We believe that the manuscript is substantially improved after making the suggested revisions.

  • Firstly, the abstract is very long and should be presented in a more concentrated form.

Response: Thanks for this comment. So, we deleted some words without prejudice substance.

 In part of Materials and methods part 2.1 is very unclear. Rows 150-151 present information about fertilizers applied but presented amounts (regarding to dimentions) are unbelievable, please check it..

Response: Thank you very much for your comment.  it was corrected (Line 149-150).

  • Row 152 presents information about weather conditions but such form is null and void as there is no real information available, an additional file with meteorological information could be provided.

Response: Thank you for your comment. We added table S1 about weather conditions in supplementary data (Line 152).

  • Tables 2, 3, 4, 6, 9 are very broad and detailed but analysing their information is tricky. Could you convert come of them to figures or convert to tables with more concentrated information. Many traits have been analysed, but it should also be valuable to provide practical options on which traits can be practically applied to assess hundreds or thousands of genotypes. I also suggest that the manuscript could be shortened by analysing the most important features and mentioning the less valuable ones briefly, or that some of the information could be moved to the supplementary files.

 Response: Thank you very much for your valuable comments. In fact, after my readings for this comment, I checked the manuscript in an attempt to shorten it, but he found that all the information was needed Because it's sequenced on each other. Also, I found other manuscripts that were close to our manuscript such as(doi.org/10.1111/plb.13271 , doi:10.3390/agronomy5020200, doi.org/10.3390/plants10050931, doi.org/10.3389/fpls.2015.00374 .)

Reviewer 2 Report

In order to cope with the challenge of climate warming, the selection of heat-resistant crop varieties has become increasingly important. Different approaches were explored in this study including agro-physiologic indices and multidimensional analyses for detecting heat tolerance in wheat genotypes, which can provide important references for wheat heat-resistant screening.  

However, my biggest concerns lie in the following aspects.

(1) Too few samples were selected for this study and there is no evidence that these samples are sufficiently genetically and geographically representative to explore such methods.

(2) In this study, two growing seasons were conducted to avoid the influence of climate differences in different years. This is good. However, the important influence of environment (or location) on phenotype was not considered. There is not enough evidence that the same approach works in other settings.

(3) Have the heat resistance of the selected samples been identified? The effectiveness and accuracy of the method can only be tested if the heat resistance of the material is known.

(4) The abstract needs to be streamlined and focused.

Author Response

Dear Reviewer 2

Thank you very much for the valuable comments obtained.  Kindly, find our response to the comments point by point below. Also, the changes have been made in the MS as track changes.

  • In order to cope with the challenge of climate warming, the selection of heat-resistant crop varieties has become increasingly important. Different approaches were explored in this study including agro-physiologic indices and multidimensional analyses for detecting heat tolerance in wheat genotypes, which can provide important references for wheat heat-resistant screening. However, my biggest concerns lie in the following aspects.

 Response: We greatly appreciate your critical observations as well as your contribution, and constructive and helpful comments to improve MS. We hope that we could address your comments by revisions made in the manuscript. We believe that the manuscript is substantially improved after making the suggested revisions.

  • (1) Too few samples were selected for this study and there is no evidence that these samples are sufficiently genetically and geographically representative to explore such methods.

Response: Thank you very much for the great and valuable comment. We agree with the Reviewer that no evidence that these samples are sufficiently genetically and geographically, so our manuscript highlights the importance and role of statistical analysis to address these points. Therefore, we used the tolerance index that is obtained via comparing the performance of the genotype under control to that under heat stress, giving rise to an index from 0 to 100. This was followed by multivariate analysis techniques and multidimensional methods in order to detect the samples’ genetic and geographical representation at high accuracy. Concerning the number of genotypes used we think they are not few since previous literatures have frequently used less sample size than that was applied in our study (doi.org/10.1111/plb.13271 , doi:10.3390/agronomy5020200, doi.org/10.3389/fpls.2015.00374, doi: 10.1080/17429145.2019.1603406, doi:10.3390/agronomy9040211, doi.org/10.1016/j.eja.2004.03.002,  doi.org/10.3390/plants10112549).

  • (2) In this study, two growing seasons were conducted to avoid the influence of climate differences in different years. This is good. However, the important influence of environment (or location) on phenotype was not considered. There is not enough evidence that the same approach works in other settings.

Response: Thank you for your comment. Test of the homogeneity of error variance of heat tolerance index between two seasons showed non-significant, so we used combined data of two seasons. And these results are for our study and they may vary with another study.

  • (3) Have the heat resistance of the selected samples been identified? The effectiveness and accuracy of the method can only be tested if the heat resistance of the material is known.

Response: Thank you for your comment. The heat resistance of the selected genotypes is recognized using the tolerance index, that is obtained via comparing the performance of the genotype under control to that under heat stress, followed by using cluster analysis that relies on the genetic dissimilarity matrix between genotypes (Euclidean distance and Ward's method of agglomeration) in the five major tolerant categories (high tolerant, tolerant, moderate, sensitive and highly sensitive).

  • (4) The abstract needs to be streamlined and focused.

Response: Thanks for this comment. So, we deleted some words without prejudice substance.

Reviewer 3 Report

General:

This paper is nicely drafted and easy to read.

Abstract:

Line 13: replace seriously with significantly. Seriously is not a scientific word

Introduction:

Line 48: witnessed a cascading impact of climate change…

Line 49: agricultural productivity

Line 52: land degradation

Line 67: Djanaguiraman et al [8]

Line 69: times and places

Line 73: optimal range damages

Line 75: Please see recent paper and cite:  Yadav, M. R., Choudhary, M., Singh, J., Lal, M. K., Jha, P. K., Udawat, P., ... & Prasad, P. V. (2022). Impacts, Tolerance, Adaptation, and Mitigation of Heat Stress on Wheat under Changing Climates. International Journal of Molecular Sciences, 23(5), 2838.  

Line 91: delete ‘a’

Line 109: to achieve accuracy in

M&M:

Line 150 and elsewhere: m-2, please keep it as superscript (all units and chemical formula’s subscript: p2o5)

Results:                                                                                                                                         

Line 300: significantly different can be shown be symbol or letters

Line 323: Please improve figure quality of PCA. All figures quality should be improved

Discussion:

I would suggest the authors to have a more supported discussion with references considering the main point: The limitations of method and considerations when to apply the studied methodology and then the potential next steps or further investigation to address these limitations.

References: Please double check the style of references and missing one

Author Response

Dear Reviewer 3

Thank you very much for the valuable comments obtained.  Kindly, find our response to the comments point by point below. Also, the changes have been made in the MS as track changes.

  • General: This paper is nicely drafted and easy to read.

Response: We greatly appreciate your critical observations as well as your contribution, and constructive and helpful comments to improve MS. We hope that we could address your comments by revisions made in the manuscript. We believe that the manuscript is substantially improved after making the suggested revisions.

  • Abstract: Line 13: replace seriously with significantly. Seriously is not a scientific word

Response: Thanks for this observation. In Line 13 was replaced seriously with significantly

  • Introduction: Line 48: witnessed a cascading impact of climate change… Introduction: Line 48: witnessed a cascading impact of climate change…

Response: Thanks for this comment. In Line 48 was corrected

  • Line 49: agricultural productivity

Response:  Thanks for this comment. In Line 49 was corrected

  • Line 52: land degradation

Response:  Thanks for this comment. In Line 52 was corrected

  • Line 67: Djanaguiraman et al [8]

Response:  Thanks for this comment. In Line 67 was corrected

  • Line 69: times and places

Response:  Thanks for this comment. In Line 69 was corrected

  • Line 73: optimal range damages

Response:  Thanks for this comment. In Line 73 was corrected

  • Line 75: Please see recent paper and cite: Yadav, M. R., Choudhary, M., Singh, J., Lal, M. K., Jha, P. K., Udawat, P., ... & Prasad, P. V. (2022). Impacts, Tolerance, Adaptation, and Mitigation of Heat Stress on Wheat under Changing Climates. International Journal of Molecular Sciences, 23(5), 2838.

Response:  Thanks for this comment. We added in Line 75.

  • Line 91: delete ‘a’

Response:  Thanks for this comment. We deleted ‘a’ in Line 91

  • Line 109: to achieve accuracy in

Response:  We added in Line 109.

  • Materials and Methods
  • Line 150 and elsewhere: m-2, please keep it as superscript (all units and chemical formula’s subscript: p2o5)

Response:  Line 150-154: modified was done as recommended.

  • Results
  • Line 300: significantly different can be shown be symbol or letters

Response:  Thanks for this comment. addition of any symbol or letters will change format the table (line 299).

  • Line 323: Please improve figure quality of PCA. All figures quality should be improved

          Response:  Thanks for this comment. We improved them.

  • Discussion
  • I would suggest the authors to have a more supported discussion with references considering the main point: The limitations of method and considerations when to apply the studied methodology and then the potential next steps or further investigation to address these limitations.

Response: Thanks for this comment. Improvements have been made (Line 586-592). The studied wheat genotypes exhibited wide variation in heat tolerance. The stress tolerance index, typically used as a criterion to evaluate tolerance, has been used for screening genotypes multiple recent literatures [29,35,45,52]. They have relied on quite fewer genotypes compared to our study. Although the number of applied genotypes could sound insufficient, it gave rise to reliable selection criteria and accurate results when multivariate analysis techniques and multidimensional methods were conducted for discriminating their heat tolerance[29,52-54].

  • References: Please double check the style of references and missing one.

Response: Thanks for this comments. The review was made .

Round 2

Reviewer 2 Report

The authors addressed all my concerns except the environment (location) effect to the results. This may affect the wide adaptability for the methods. The authors also acknowledge that the results may vary in other studies. Therefore, it would be better to increase some relevant content in the discussion.

Author Response

Dear Academic Editor of agronomy 
Thank you very much for the valuable comments obtained from the Editor and Reviewers of Agronomy Journal. Kindly, find our response to the comment below. Also, the changes have been made in the MS as track changes.  
# Reviewer 2
    The authors addressed all my concerns except the environment (location) effect to the results. This may affect the wide adaptability for the methods. The authors also acknowledge that the results may vary in other studies. Therefore, it would be better to increase some relevant content in the discussion.

Response: We greatly appreciate your critical observations as well as your contribution, and constructive and helpful comments to improve MS. We hope that we could address your comments by revisions made in the manuscript. We believe that the manuscript is substantially improved after making the suggested revisions. In lines (712-720) we added the part about your comment "Since heat-tolerant trait is very complicated and affected by many genetic and environmental factors and their interactions, an overall understanding of the genetic basis and plant responses to this stress and their interaction with the environment is needed [6,73-75]. Recently discovered new statistical methods, we would be using in a future study with more the environments such as the multi-trait genotype-ideotype distance index (MGIDI), the weighted average of absolute scores (WAASB) index, and a superiority index that allows weighting between mean performance and stability (WAASBY), helping breeders to take appropriate decisions when selecting genotypes are more stable and commended in multi-environment trials [74-76]''